# Trace Reconstruction for DNA Data Storage using Language Models

## Abstract

DNA is a promising storage medium due to its high information density and longevity. However, the storage process introduces errors, thus algorithms and codes are required for reliable storage. A common important step in the recovery of the information from DNA is trace reconstruction. In the trace reconstruction problem, the goal is to construct a sequence from noisy copies corrupted by deletion, insertion, and substitution errors. In this paper, we propose to use language models trained with next-token prediction for trace reconstruction. A simple channel model for the DNA data storage pipeline allows for self-supervised pretraining on large amounts of synthetic data. Additional finetuning on real data enables us to adapt to technology-dependent error statistics. The proposed method (TReconLM) outperforms state-of-the-art trace reconstruction algorithms for DNA data storage, often recovering significantly more sequences.

## 1 Introduction and Motivation

An important problem in DNA data storage and biological data analysis is trace reconstruction: Given multiple versions of a string (traces) corrupted by deletions, insertions, and substitutions, the goal of trace reconstruction is to reconstruct the original string from as few traces as possible.

Within DNA data storage, the string to be reconstructed is typically a DNA sequence consisting of 50-200 bases adenine (A), cytosine (C), guanine (G), and thymine (T). The traces are corrupted by deletions, insertions, and substitutions through the writing, storage, and reading processes. Trace reconstruction is often used as an important information reconstruction step (Antkowiak et al., 2020; Organick et al., 2018; Bar-Lev et al., 2024).

The current trace reconstruction algorithms used for DNA data storage, including general trace reconstruction algorithms like MUSCLE (Edgar, 2004) and ITR (Sabary et al., 2020), and algorithms specialized for DNA data storage like RobuSeqNet (Qin et al., 2024) do not perform well for high error rates and a small number of traces. Therefore, it is of interest to develop trace reconstruction methods that can operate in the medium to high noise regime for a small number of traces, typically two to ten traces. Being able to perform trace reconstruction from few traces can further improve the reliability of current DNA data storage systems. Furthermore, some classical algorithms (Viswanathan & Swaminathan, 2008; Srinivasavaradhan et al., 2021) utilize estimated values for the error probabilities but do not consider the distribution of these over the sequence length, others operate solely on the observed sequences (Sabary et al., 2020; Edgar, 2004; Gopalan et al., 2018). This motivates the use of data-driven approaches.

In this work, we treat trace reconstruction for DNA data storage as a next-word prediction problem. We train language models to predict sequence estimates from noisy observations.

Our contributions are as follows:

- We develop a language model-based trace reconstruction method called **TReconLM**, standing for Trace Reconstruction with a Language Model. The method outperforms state-of-the-art trace reconstruction methods for reconstructing DNA sequences from few traces. Our method is trained on synthetic data, overcoming the lack of available real data.
- We demonstrate that additional finetuning on real datasets can overcome the distribution shift to real systems, and can benefit data reconstruction in DNA storage systems.

## 2    RELATED WORK

Theoretical works on trace reconstruction typically ask for the minimum number of traces (obtained through a deletion channel) such that a binary string can be reconstructed with high probability (Batu et al., 2004; Holenstein et al., 2008; De et al., 2017; Holden & Lyons, 2020; Chase, 2021). However, perfect reconstruction using only a few traces is usually not possible.

Several trace reconstruction algorithms suitable for DNA data storage applications have been developed. For traces with deletions, Batu et al. (2004) introduced the bitwise majority alignment (BMA) algorithm, which is based on symbol-wise majority voting. Viswanathan & Swaminathan (2008) extended the algorithm to traces with deletions, insertions, and substitutions. Gopalan et al. (2018) provide another BMA-based method.

In the work of Antkowiak et al. (2020), the trace reconstruction was achieved by first performing a multiple sequence alignment (MSA) using the MUSCLE algorithm (Edgar, 2004) followed by a majority vote of each column of the alignment.

Sabary et al. (2020) proposed several reconstruction methods for DNA data storage based on dynamic programming, namely shortest common supersequence and longest common subsequence algorithms. The proposed iterative algorithm (ITR) gives state-of-the-art performance. Srinivasavaradhan et al. (2021) introduced the TrellisBMA algorithm, which combines the BCJR algorithm (Bahl et al., 1974) and BMA-based algorithms.

Qin et al. (2024) proposed a neural network-based method (RobuSeqNet) for trace reconstruction using a combination of an attention module, conformer-encoder, and LSTM-decoder. The attention module assigns a score to each sequence within a cluster which allows to reduce the influence of erroneously clustered sequences. Sequences are one-hot encoded and padded to a fixed length to obtain a matrix representation. These matrix representations are then added for different sequences. For perfectly clustered data, the performance is slightly worse than that of the iterative algorithm proposed in Sabary et al. (2020).

Bar-Lev et al. (2024) proposed an end-to-end solution for DNA data storage (DNAformer), including a coding scheme. This involves using transformers for trace reconstruction. The neural network architecture differs from that of this work in several key aspects. First, like in Qin et al. (2024), sequences are one-hot encoded and padded to a fixed predetermined length. Second, the network consists of two branches with shared weights, where one branch operates on the reversed data. Third, the network uses an alignment module to learn the required alignment of all reads. The alignment module is followed by a transformer block without positional embeddings and causal attention masks. The method uses dynamic programming methods to postprocess the neural network outputs. The DNAformer performs similarly to ITR (Sabary et al., 2020).

In Nahum et al. (2021), a method for single-read trace reconstruction utilizing the transformer architecture was proposed where noisy sequences are grouped based on their length, and a separate transformer network is applied to each group. The method operates on a set of 256 codewords.

Finally, Dotan et al. (2023) introduced BetaAlign, an encoder-decoder-based transformer network for multiple sequence alignment of biological sequences.

## 3    BACKGROUND ON DNA DATA STORAGE AND PROBLEM STATEMENT

DNA data storage is an interesting storage technology because of its high information density and longevity. For technological constraints, it is currently not possible to write long sequences of DNA, and therefore data in DNA is stored on many relatively short sequences, i.e., sequences of length $L = 50$ to $L = 200$ bases.

The binary data to be stored is therefore mapped with an encoder to a set of sequences $\mathcal{D} = \{\boldsymbol{x}_1, \ldots, \boldsymbol{x}_M\}$, where $\boldsymbol{x}_i \in \{A, C, T, G\}^L$ is a vector over the alphabet consisting of the four bases: adenine (A), cytosine (C), guanine (G), and thymine (T).

The set of sequences $\mathcal{D}$ is then written in DNA (synthesized) and stored. After reading (sequencing), we obtain many unordered erroneous reads. The first step in recovering the information often involves a clustering step to group sequences that are potentially perturbed versions of the same orig-

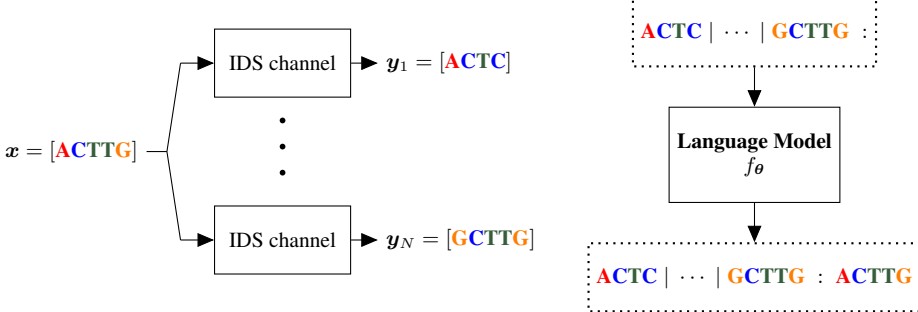

Figure 1: **Left panel:** The goal of trace reconstruction is to reconstruct a sequence $x$ from $N$ noisy copies $y_1, \ldots, y_N$ independently corrupted by insertions, deletions, and substitutions (IDS). **Right panel:** We propose to perform trace reconstruction by formulating trace reconstruction as a next-word prediction problem, and training a transformer $f_\theta$ to perform this task.

inal sequence $x_i$. In general, we can have multiple clusters for sequences originating from the same original sequence $x_i$ (Antkowiak et al., 2020; Organick et al., 2018; Rashtchian et al., 2017), and clusters can in principle also contain sequences that are perturbed versions of two or more original sequences.

Given the clusters, trace reconstruction is performed to compute cluster-wise sequence estimates $\hat{x}_i$ of the original sequences $x_i$ which reduces the error rates. Finally, a decoder relying on error-correcting codes picks up the remaining errors and reconstructs the information.

In this paper, we consider the trace reconstruction problem in the context of DNA data storage: Given $N$ noisy observations $y_1, \ldots, y_N$ of a (DNA) sequence $x$ corrupted by unknown deletions, insertions, and substitutions, the goal is to estimate the original sequence. See Figure 1, left panel, for an illustration. We assume the original sequence $x$ to consist of bases chosen uniformly at random, since several DNA data storage systems use pseudorandom sequences to randomize the bases within each of the sequences (Antkowiak et al., 2020; Organick et al., 2018).

## 4 METHOD

We formulate trace reconstruction as a next-word prediction problem and train a sequence-to-sequence model (we consider transformers) on this next-word prediction problem. Given a set of $N$ sequences $\mathcal{C} = \{y_1, \ldots, y_N\}$, our goal is to train a model $f_\theta$ with parameters $\theta$ so that if we prompt it with the concatenation of the observations

$$p = y_1 \mid y_2 \mid \ldots \mid y_{N-1} \mid y_N :, \tag{1}$$

then the model provides an estimate $\hat{x}$ of the original sequence $x$ as a completion for this prompt. Here, we introduced the "|" token to concatenate the noisy reads and the ":" token to mark the end of the noisy observations. Thus, the vocabulary of our model is given by $\mathcal{V} = \{\text{A}, \text{C}, \text{T}, \text{G}, \text{"|"}, \text{":"}\}$.

The model, prompted with $p$, generates the sequence estimate $\hat{x}$ by predicting the $L$ tokens following the prompt in an autoregressive manner via multiple forward passes. The generation is performed via greedy sampling, where we choose the most likely token in each prediction step. See Figure 1, right panel, for an illustration of the method.

### 4.1 PRETRAINING AND PRETRAINING DATA GENERATION

In DNA data storage, each sequence is corrupted by deletions, insertions, and substitutions (Heckel et al., 2019). As stated before, the original sequence $x$ can be assumed to consist of bases chosen uniformly at random.

Therefore, we generate training data as follows. We first generate an original sequence $x \in \{\text{A}, \text{C}, \text{G}, \text{T}\}^L$ of length $L$ uniformly at random, and then obtain associated noisy observations

$\boldsymbol{y}_1, \ldots, \boldsymbol{y}_N$ by independently corrupting the sequence $\boldsymbol{x}$ with deletions, insertions, and substitutions.

Specifically, to generate the noisy observation $\boldsymbol{y}_j$ we go through the original sequence $\boldsymbol{x}$ element-wise and introduce independent deletions, insertions, substitutions, or transmissions (i.e., no change) with probabilities $p_I$, $p_D$, $p_S$, and $p_T$ respectively. Note that $p_I + p_D + p_S + p_T = 1$. Figure 2 depicts the transition from $x_\ell$ to $x_{\ell+1}$. We then concatenate the noisy observations $\boldsymbol{y}_1, \ldots, \boldsymbol{y}_N$ together with the original target sequences to obtain one training instance

$$\boldsymbol{y}_1 \mid \boldsymbol{y}_2 \mid \ldots \mid \boldsymbol{y}_{N-1} \mid \boldsymbol{y}_N : \boldsymbol{x}. \tag{2}$$

We generate large numbers of such training examples (see the experiments Section 5.2 for details), and vary the number of traces $N$ in the range from two to ten, since this is a practically relevant and challenging regime. With regards to the error probabilities, we uniformly sample the error probabilities $p_I$, $p_D$, and $p_S$ from the interval $[0.01; 0.1]$ for each training instance.

On the so-generated training data, we train the transformer model by minimizing the cross entropy loss between the predicted original sequence, $\hat{\boldsymbol{x}}$, and the original sequence, $\boldsymbol{x}$.

## 4.2 FINETUNING FOR REAL DATA

Our pretraining data is generated by varying the deletion, insertion, and substitution errors in some range, and by inserting deletions, insertions, and substitutions independently at each position of the original sequence. We also generate each noisy sequence independently of the other.

However, the data in practice is obtained differently in that there are specific deletion, insertion, and substitution error probabilities, and the independence assumption is violated. For example, the error probabilities can depend on the position within a string (Antkowiak et al., 2020).

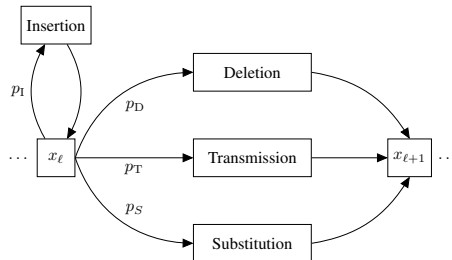

Figure 2: IDS channel

Thus, there is a distribution shift between our training data generation and the data we wish to apply our method to in practice. To overcome this mismatch, we can finetune a pretrained model on real-world data, as discussed in our real-world experiments in Section 5.3.1 and 5.3.2, where we consider two publicly available datasets, namely the Noisy-DNA dataset (Antkowiak et al., 2020) and the Microsoft dataset (Srinivasavaradhan et al., 2021). From these datasets, we extract ground-truth sequences $\boldsymbol{x}$ and associated noisy traces $\boldsymbol{y}_1, \ldots, \boldsymbol{y}_N$ and construct training examples of the form given in Equation 2. We then finetune on these analogously as we do pretraining.

It is also possible to finetune or train the model directly on simulated data matching the respective channel characteristics as closely as possible.

## 5 EXPERIMENTS

In this section, we evaluate the performance of our proposed language model-based trace reconstruction method, TReconLM, on synthetic data and on real DNA storage data. We find that our proposed approach based on next-word prediction outperforms state-of-the-art trace reconstruction methods for DNA storage, specifically ITR (Sabary et al., 2020), DNAformer (Bar-Lev et al., 2024) and RobuSeqNet (Qin et al., 2024). We evaluate performance with the following metrics:

- The Hamming distance $d_H(\boldsymbol{x}, \hat{\boldsymbol{x}})$ between the original sequences $\boldsymbol{x}$ and the reconstructed one, $\hat{\boldsymbol{x}}$, which is the number of positions where the sequences $\boldsymbol{x}$ and $\hat{\boldsymbol{x}}$ differ, normalized by the sequence length. For this distance measure, we slightly postprocess by random filling of sequence estimates shorter than $L$ and cutting sequence estimates longer than length $L$ to length $L$.

- The Levenshtein distance $d_L(\boldsymbol{x}, \hat{\boldsymbol{x}})$, which is the minimum number of single-character edit-events (deletions, insertion, and substitutions) to transform the sequence estimate $\hat{\boldsymbol{x}}$ into the ground-truth vector $\boldsymbol{x}$, again normalized by the sequence length $L$.

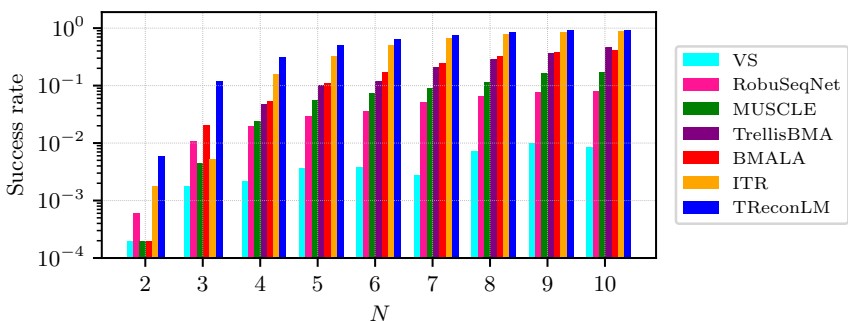

Figure 3: Success rates for synthetically generated in-distribution data ($L = 60$) for different cluster sizes $N$. Our reconstruction method TReconLM is able to recover more sequences than any of the baseline methods.

- The success rate, which is the fraction of error-free reconstructed vectors, i.e., $SR(\boldsymbol{x}, \hat{\boldsymbol{x}}) = \frac{\#d_{\mathrm{H}}(\boldsymbol{x}, \hat{\boldsymbol{x}})=0}{\#\text{test instances}}$.

The Hamming distance captures the positional accuracy, and the Levenshtein distance measures overall string similarity.

We consider decoder-only transformers (Radford et al., 2019).

## 5.1 BASELINES

We compare our proposed trace reconstruction technique to two types of trace reconstruction methods, dynamic programming-based and deep learning-based methods.

We consider the following dynamic programming-based algorithms: The iterative algorithm (ITR) (Sabary et al., 2020), trace reconstruction using MUSCLE (Edgar, 2004) followed by majority voting, and the TrellisBMA algorithm (Srinivasavaradhan et al., 2021). Furthermore, we compare to BMALA (Gopalan et al., 2018) and VS (Viswanathan & Swaminathan, 2008). Due to the prohibitively long running time of the TrellisBMA algorithm, we show reconstruction results only for one experiment on synthetic and one real data experiment.

As a neural network-based reconstruction method, we consider RobuSeqNet (Qin et al., 2024) and DNAformer (Bar-Lev et al., 2024). We also compare to GPT4oMini in Appendix C.2.

## 5.2 EVALUATION ON SYNTHETIC DATA

We first evaluate the trace reconstruction performance on synthetic data generated synthetically like our training data for three sequence lengths $L$, namely 60, 110, and 180 bases. In general, training a separate model for each trace reconstruction problem with number of sequences $N$ gives the best performance. However, to have one model that is applicable to different values of the number of traces $N$, we train one model for the reconstruction of two to five sequences and another model for six to ten traces. We train transformer models with 300M parameters on 32M training examples and evaluate on 5000 random test examples, generated equally as the training data.

Success rates for $L = 60$ are in Figure 3, where we see that TReconLM outperforms the baseline methods for each cluster size. It also outperforms RobuSeqNet, a neural network based method. However RobuSeqNet is a much smaller model, but when controlling for the model size, TReconLM also outperforms RobuSeqNet, as can be seen in Appendix C.1. Figure 4 shows the average Hamming and Levenshtein distance for different cluster sizes, and our language model-based reconstruction gives the best results in both metrics.

The results for $L = 110$ and $L = 180$ can be found in Appendix A and show that also for longer sequences, our reconstruction method outperforms the baseline methods on synthetic data.

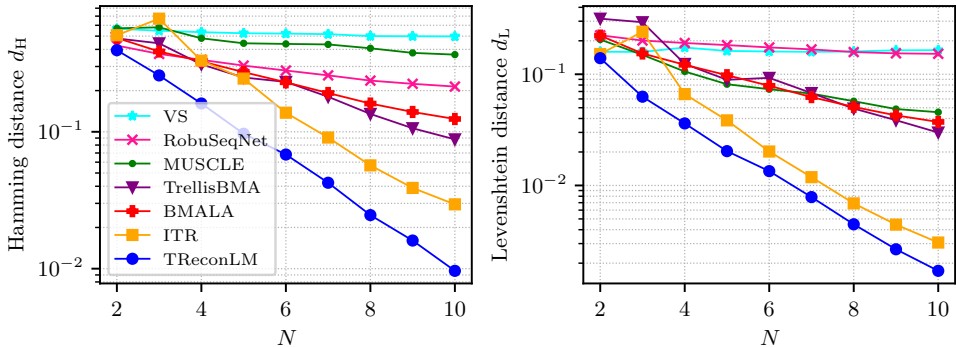

Figure 4: Hamming and Levenshtein distance for IDS data and sequence length $L = 60$. We see that TReconLM gives the best overall result for all cluster sizes. While the curves for ITR have similar slopes, we can observe a clear performance gap across cluster size of three to ten.

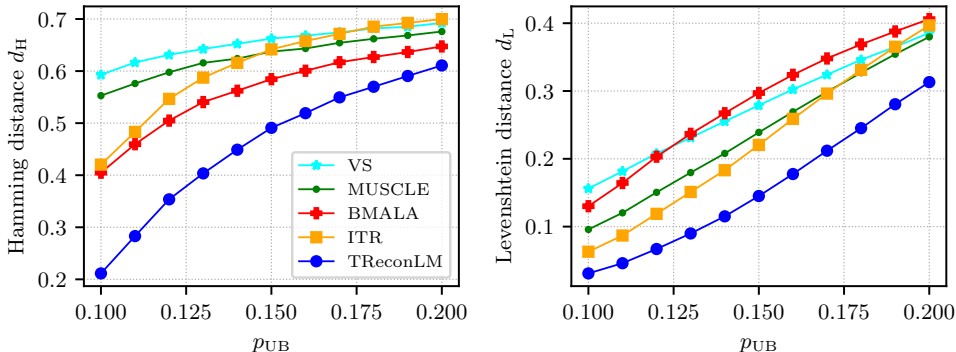

Figure 5: Noise sweep for the IDS channel. We show Hamming and Levenshtein distances across different values of the upper noise bound $p_{\text{UB}}$. Our reconstruction model generalizes to higher noise values outperforming all baseline methods.

### 5.2.1 GENERALIZATION TO HIGH NOISE VALUES

To test TReconLM's generalization capabilities, we evaluate the trace reconstruction performance for high noise values, which were not used during the pretraining process. For the error probabilities $p_{\text{I}}$, $p_{\text{D}}$, and $p_{\text{S}}$, we gradually increase both the lower and the upper bound ($p_{\text{LB}}$ and $p_{\text{UB}}$) of the uniform noise distribution $\mathcal{U}[p_{\text{LB}}; p_{\text{UB}}]$ by 0.01 at the same time. For a cluster size of $N = 4$ and a sequence length of $L = 110$, we evaluate a set of 5000 random sequences for each noise interval. Figure 5 shows that TReconLM generalizes to higher noise values, outperforming the baseline trace reconstruction methods, even under this mismatch of training- and test-data. As before, the model consists of 300M parameters and is trained on 32M training instances.

### 5.3 EXPERIMENTS ON REAL DATA

In this section, we finetune pretrained transformer models on real-world data, in order to quantify the benefits that our proposed method yields for DNA storage applications. We consider the Noisy-DNA dataset provided by Antkowiak et al. (2020), which uses a technology for writing that is comparatively cost-efficient but induces many errors. Second, we consider the Microsoft dataset (Srinivasavaradhan et al., 2021), which was obtained using nanopore sequencing, which again induces many errors. Thus, for both scenarios reconstruction is very challenging, and trace reconstruction is currently used.

By using the data from one storage experiment for finetuning, we obtain a technology-adapted model that can be employed in any subsequent experiment with the same sequence length $L$ and reading/writing equipment.

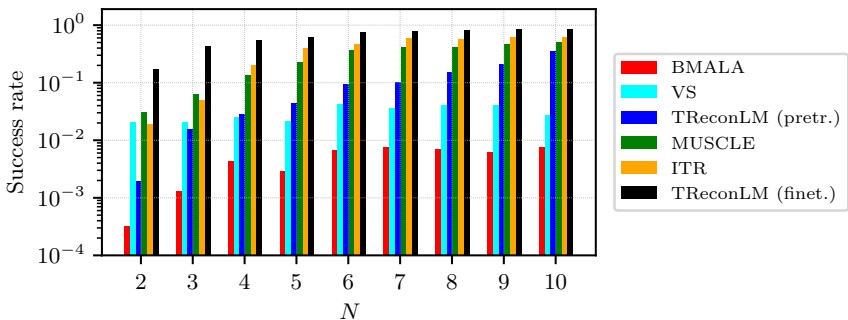

Figure 6: Success rates for Noisy-DNA dataset. The pretrained model is not able to overcome the mismatch between the IDS channel and the error statistics of the Noisy-DNA dataset. A small finetuned model with 20M parameters achieves the best success rate over all considered cluster sizes.

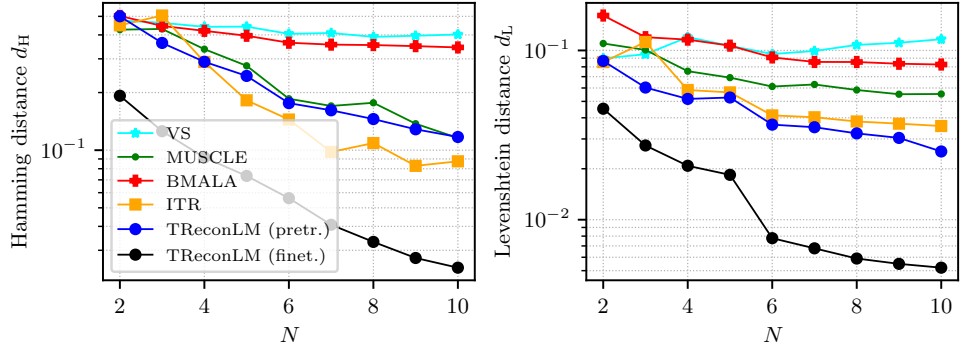

Figure 7: Hamming and Levenshtein for Noisy-DNA dataset. The pretrained model fails to outperform the iterative algorithm (ITR) in Hamming distance. Finetuning gives a substantial reduction in both Hamming and Levenshtein distance.

### 5.3.1 REAL DATA EXPERIMENT 1: NOISY-DNA DATASET

The Noisy-DNA dataset contains $M = 16383$ ground-truth sequences, each of length $L = 60$ bases, along with unclustered noisy reads. We first discard reads outside the interval of $[55, 70]$ which yields $1.4e7$ unclustered noisy observations. Each sequence contains a unique index of length twelve, which can be used to order the sequences for data reconstruction. The error probabilities were estimated in Antkowiak et al. (2020) with $p_I = 0.057$, $p_D = 0.06$, and $p_S = 0.026$, which is high. The error probabilities depend on the position within the sequence (Antkowiak et al., 2020). The insertion probability is up to $p_I = 0.3$ towards the sequence end.

We consider two pretrained models consisting of 20M parameters. The first model is trained to reconstruct two to five sequences, and the second model to reconstruct six to ten. Both are trained on 32M instances of form shown in Equation 2. Since we have two pretrained models that we want to finetune, we generate a separate training/validation and testing set for each model. In order to obtain training/validation and test sets, we first cluster all reads by the sequence index, which yields $M$ index-clusters that are separated into train/validation and test index-clusters. For the first model (two to five sequences), we generate a test set out of 500 index-clusters and use the remaining index-clusters for the generation of a training and validation set. For the second model (six to ten sequences), we use 1000 index-clusters for test set generation.

The training/validation index-clusters are processed as follows. To remove erroneously clustered sequences, we filter each of the index-clusters for training and validation by calculating the Levenshtein distance to the corresponding ground-truth sequence. Note that a sequence length of $L = 60$ gives around $60 * 0.057 + 60 * 0.06 + 60 * 0.026 = 8.58$ edit operations per noisy observation. Given this value, we discard noisy reads with Levenshtein distance smaller than five and larger than 13 to match the error probabilities stated above.

The test index-clusters are processed as follows to evaluate close to the pipeline of clustering followed by trace reconstruction used in practice. We join all reads from the test index-clusters into one set of reads. This set of reads is clustered using the algorithm proposed in Zorita et al. (2015), which

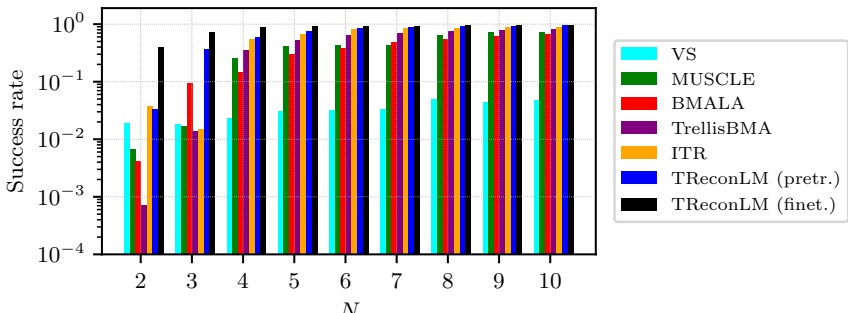

Figure 8: Success rates for Microsoft dataset. Pretrained and finetuned models give the best overall performance.

operates based on sequence similarity and was already employed in previous DNA data storage experiments (Organick et al., 2018). In order to evaluate the performance of different reconstruction algorithms, we find the ground-truth sequence that has the lowest average Levenshtein distance for each test cluster.

The filtered training/validation index-clusters and the test clusters (obtained through Zorita et al. (2015)) are split into subclusters of either size two to five or six to ten in order to generate training and test examples according to Equation 2. Finally, for the model that is finetuned on two to five sequences, we have a training/validation set consisting of about 260000 instances and a test set with 3000 examples. For the second model, the associated training/validation set has 127000 training instances and about 2000 test examples. For both training/validation sets we use 90% for training and the remaining 10% for validation.

Due to the high relative frequency of cytosine at the end of most reads, caused by the insertion probability of $p_{\mathrm{I}} = 0.3$ towards the end of the reads, we employ preprocessing of all test reads for the baseline methods and the pretrained transformer models by removing any trailing C bases. This increases the overall performance of the baselines. For the finetuned models, which learn the error statistics of the Noisy-DNA dataset, preprocessing is not needed.

Figure 6 shows the success rates for different cluster sizes. We observe that the pretrained model is not able to generalize to the technology-dependent error statistics. However, by finetuning we obtain a significant performance increase where we are able to recover one order of magnitude more sequences than the iterative algorithm (ITR) for cluster sizes of two and three. Figure 7 compares Hamming and Levenshtein distance. We see that the pretrained models achieve lower Levenshtein distance than the baselines but fail to do so for Hamming distance. The finetuned model outperforms all other methods by a significant margin for all cluster sizes.

### 5.3.2 REAL DATA EXPERIMENT 2: MICROSOFT DATASET

The Microsoft dataset contains $M = 10000$ ground-truth sequences with $L = 110$ and 269707 clustered noisy reads, one cluster per sequence. This results in large clusters often containing more than ten reads. The sequences were clustered using the algorithm by Rashtchian et al. (2017). The error probabilities where estimated by Srinivasavaradhan et al. (2021) to be $p_{\mathrm{I}} = 0.017$, $p_{\mathrm{D}} = 0.02$, and $p_{\mathrm{S}} = 0.022$.

Similar as for the Noisy-DNA dataset, we train and evaluate two models, one for small clusters (two to five) and one for medium values of $N$ (six to ten sequences).

Therefore, we split the large clusters into smaller subclusters. To obtain training and test data for the model trained to reconstruct sequences based on clusters of size two to five, we first split the 10000 clusters into 9500 for training/validation and 500 for testing. Both are split into subclusters of size two to five which results in 53000 training/validation examples and 2800 test examples. To obtain training and test data for the model trained to reconstruct sequences of length six to ten sequences, we use 1000 clusters of testing and the remaining 9000 for training/validation. Again, splitting both into subclusters of size six to ten gives 2700 test examples and 24000 train/validation instances. Both of the training/validation sets are split with 90% for training and 10% for validation.

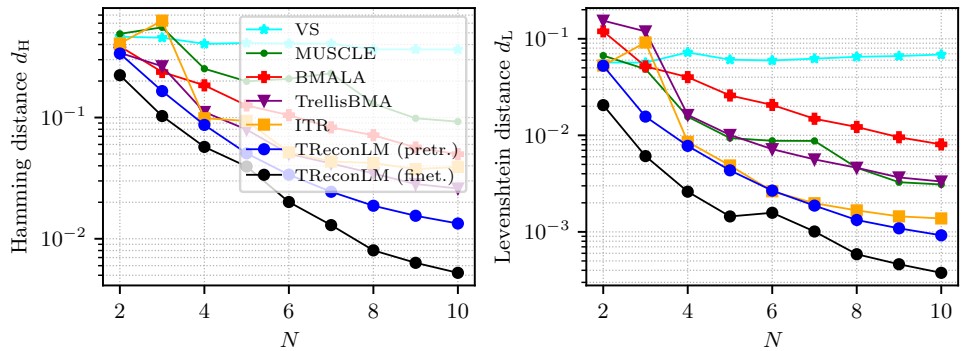

Figure 9: Hamming and Levenshtein distances for Microsoft dataset. Pretrained and finetuned models are able to outperform the considered baseline algorithms.

We then finetune 300M transformer models (see Appendix A, pretrained on 32M instances) on each of the two training sets and test on the respective test sets.

For clusters of size two, finetuning allows recovering one order of magnitude more sequences than the pretrained model. Figure 9 provides Hamming and Levenshtein distances for all considered values of the cluster size $N$. Both pretrained and finetuned models achieve better performance than the baselines, likely because the pretrained models can overcome the distribution shift between the IDS channel and the Microsoft dataset.

We now compare to DNAformer (Bar-Lev et al., 2024). Unfortunately, the implementation for the DNAformer is not available to compare to directly. However, DNAformer achieves an overall success rate of $0.8542$ on the Microsoft dataset with a 100M parameter model and dynamic programming methods for postprocessing. Since the DNAformer is trained on synthetic data we evaluate pretrained models on the Microsoft dataset. We use transformers of two model sizes, namely 20M and 300M parameters. For both cases, we consider a model for clusters of size two to five and a separate model for six to ten sequences. When clusters contain more than ten sequences, we randomly sample ten reads. The 20M parameter models achieve a success rate of $0.8508$, while the networks with 300M parameters reach $0.9242$. TReconLM achieves similar performance with a network that is five times smaller without postprocessing by any dynamic programming method and not utilizing the full cluster size.

## 5.4 SCALING LAWS FOR TRACE RECONSTRUCTION

In this section, we show preliminary results on scaling laws for the trace reconstruction problem. We train transformer models of different sizes for various numbers of tokens. We train models of sizes 3M, 10M, 21M, 37M, 85M, 170M, 300M on 8M, 16M, and 32M instances. For transformers, training compute can be estimated to be $C = 6N_P D_T$, with the number of parameters $N_P$ and the total number of training tokens $D_T$. We consider a sequence length of $L = 60$ and $N = 5$ noisy reads. Figure 10 shows the training curves for all considered models and number of training instances. It can be seen that especially for larger models, performance has not converged suggesting that TReconLM achieves better performance by further increasing compute and in particular by increasing the number of training instances relative to the model size. The corresponding experiments are running and the figure will be updated with the final runs.

## 6 CONCLUSION AND DISCUSSION

In this work, we proposed a deep learning-based method for trace reconstruction for DNA data storage. Our method achieves higher success rates and lower Hamming and Levenshtein distances than state-of-the-art trace reconstruction methods for small cluster sizes and a wide range of noise values on synthetic data as well as real-world data. Finetuning enables adaptation to the error characteristics of different synthesis and sequencing technologies, and enables to recover significantly more sequences than competing methods.

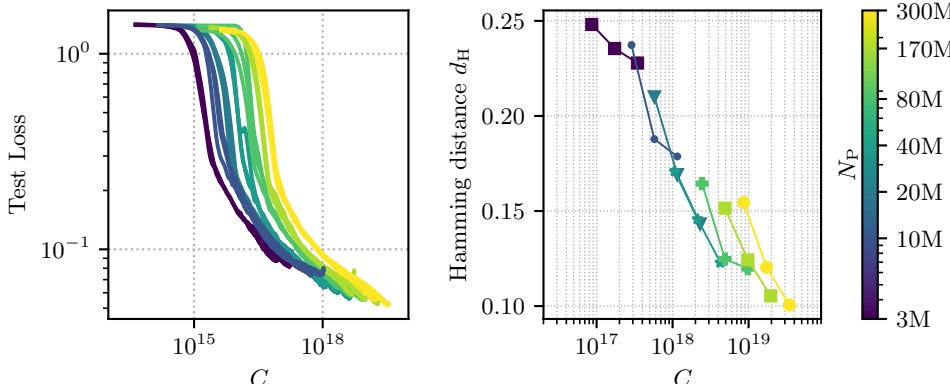

Figure 10: Preliminary scaling results for trace reconstruction. Investing further compute is expected to increase performance further, and increasing the training instances relative to the model size is also expected to give better performance.

Our proposed method is more efficient than other deep learning-based reconstruction algorithms, as networks with smaller sizes can achieve similar performance. We demonstrated that training networks with similar network sizes outperform competing data-driven approaches.

Since our method is data-driven it might not perform well if the test and train data are substantially different. In this case, finetuning can help to achieve good reconstruction results.

REPRODUCIBILITY STATEMENT

We will release all code, dataset links, and reproduction instructions on our GitHub page.

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

## A  ADDITIONAL RESULTS FOR THE IDS CHANNEL

In this section, we show additional results for the IDS channel. We consider sequence lengths of $L = 110$ and $L = 180$. Figure 11 and 12 depict success rates and distances for sequence lengths of 110 bases. We trained 300M transformer models over 32M instances, one for two to five sequences and another one for clusters of size six to ten. Similar to the results shown in Section 5.2, we can observe that our method achieves higher success rates and lower distances for all cluster sizes in the range of two to ten.

Figure 13 and 14 show similar results for $L = 180$. TReconLM is able to outperform state-of-the-art reconstruction algorithms (ITR). The results are for 300M parameter models. Due to the higher sequence length, we train on 24M instances to reduce training time.

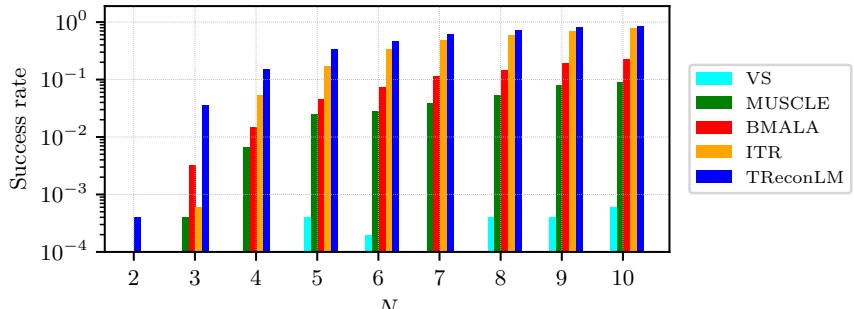

Figure 11: Success rates for IDS data and sequence length $L = 110$

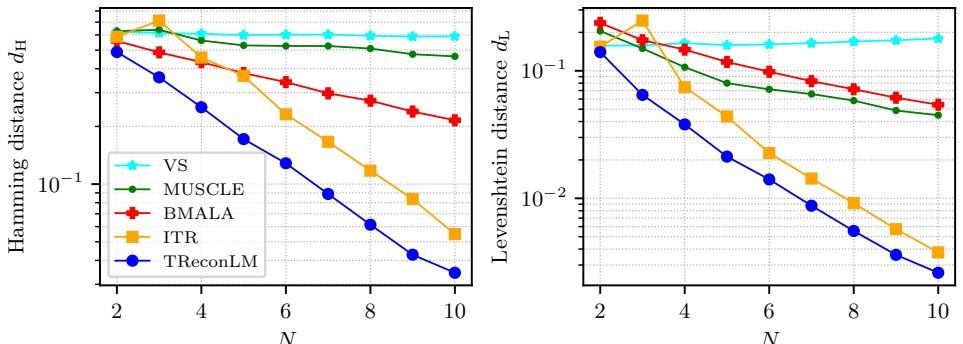

Figure 12: Hamming and Levenshtein distance for IDS data and sequence length $L = 110$

## B  MULTIPLE SEQUENCE ALIGNMENT TARGET

In this section, we evaluate different neural network targets for the trace reconstruction problem. As proposed in  Dotan et al. (2023), we can train a model $f_{\boldsymbol{\theta}}$ to learn the alignment of the observed sequences. For $N$ noisy reads $\boldsymbol{y}_1, \ldots, \boldsymbol{y}_N$, one training instance is formed as

$$\boldsymbol{y}_1 \mid \boldsymbol{y}_2 \mid \ldots \mid \boldsymbol{y}_{N-1} \mid \boldsymbol{y}_N \ : \ \mathrm{MSA}\big(\boldsymbol{y}_1, \boldsymbol{y}_2, \ldots, \boldsymbol{y}_{N-1}, \boldsymbol{y}_N\big)\#. \tag{3}$$

The vocabulary for the alignment task is given by

$$\mathcal{V}_{\mathrm{MSA}} = \{\mathrm{A, C, T, G,} \text{``}|\text{''}, \text{``:''}, \text{``-''}, \text{``\#''}\} \tag{4}$$

where we have an additional end of sequence "#" and a deletion token "-", which is used to achieve a column-wise matching of the aligned sequences. For the pretraining data generation, we know exactly the positions where a deletion, insertion, or substitution occurred, which allows us to form the correct sequence alignment. During inference, the model is provided the prompt $\boldsymbol{p}$ (Equation 1) to predict the alignment token by token until the occurrence of "#". In the next step, we write the

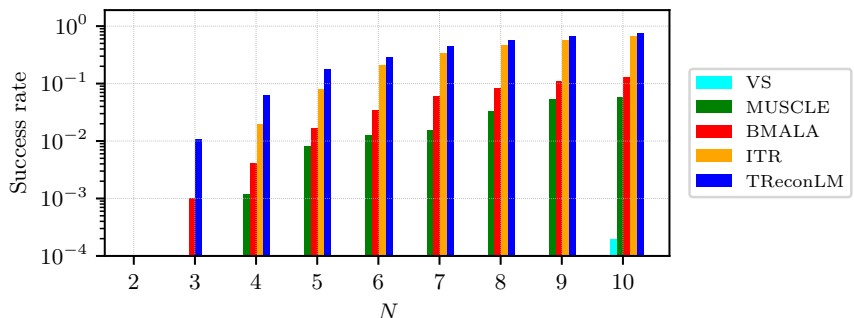

Figure 13: Success rates for IDS data and sequence length $L = 180$

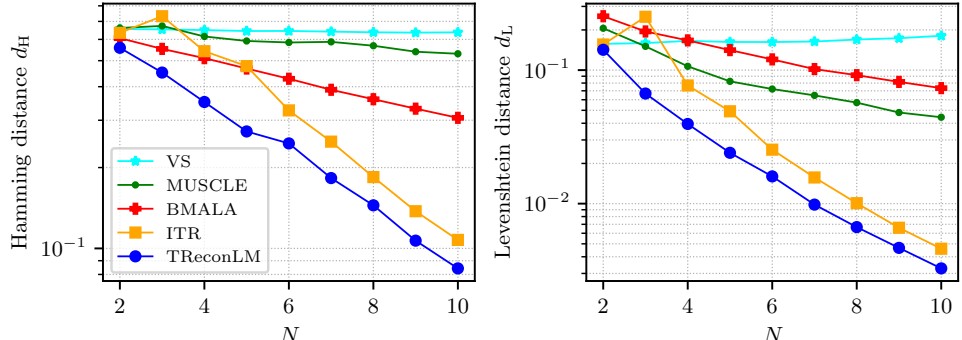

Figure 14: Hamming and Levenshtein distance for IDS data and sequence length $L = 180$

aligned sequences $\hat{\boldsymbol{y}}_1, \ldots, \hat{\boldsymbol{y}}_N$, each of length $L_{\mathrm{MSA}}$, under each other:

$$
\begin{array}{|c|c|c|c|c|}
\hline
\hat{y}_{1,1} & \hat{y}_{1,2} & \cdots\cdots & \hat{y}_{1,L_{\mathrm{MSA}}-1} & \hat{y}_{1,L_{\mathrm{MSA}}} \\
\hline
\hat{y}_{2,1} & \hat{y}_{2,2} & \cdots\cdots & \hat{y}_{2,L_{\mathrm{MSA}}-1} & \hat{y}_{2,L_{\mathrm{MSA}}} \\
\hline
\vdots & & \vdots & & \vdots \\
\hline
\hat{y}_{N,1} & \hat{y}_{1,2} & \cdots\cdots & \hat{y}_{N,L_{\mathrm{MSA}}-1} & \hat{y}_{N,L_{\mathrm{MSA}}} \\
\hline
\end{array}
\tag{5}
$$

To compute the sequence estimate $\hat{\boldsymbol{x}}$, we perform a column-wise majority vote of the alignment. The $j$-th entry of the estimated sequence $\hat{\boldsymbol{x}}$ can be calculated as

$$
\hat{x}_j = \underset{a \in \{\mathrm{A,C,T,G}\}}{\arg\max} \sum_{i=1}^{N} \mathbf{1}(\hat{y}_{i,j} = a),
\tag{6}
$$

with the indicator function $\mathbf{1}(\cdot)$. In Figure 15 we evaluate the following targets for the trace reconstruction: candidate prediction (CPRED) as described in Section 4, the MSA target as given in Equation 3 and a NESTED alignment target, where we perform a token-wise nesting of the ground-truth alignment $\mathrm{MSA}(\boldsymbol{y}_1, \ldots, \boldsymbol{y}_N)$. We also evaluate MUSCLE to compare neural network-based alignment to dynamic programming-based alignment. Figure 15 shows distances for all targets, where we can observe that the candidate prediction gives the best overall result. Furthermore, the alignment target requires higher block lengths of the transformer models compared to the CPRED target. Also, finetuning, as described in Section 4.2, is not possible because the ground-truth alignment for real data is not known in general.

## C  ADDITIONAL COMPARISONS

Here, we provide additional comparisons to RobuSeqNet as well as to GPT4oMini.

### C.1  ROBUSEQNET

In this section, we compare the performance of TReconLM to RobuSeqNet (Qin et al., 2024). The network architecture proposed in Qin et al. (2024) is rather small with about 2.5M parameters and

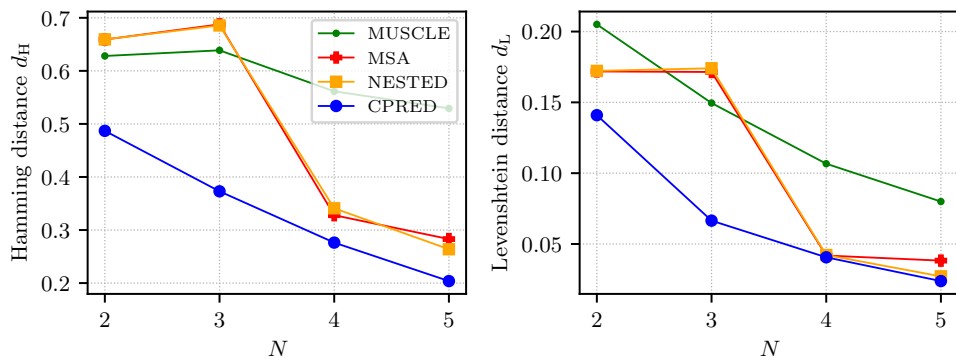

Figure 15: Comparison of different neural network targets. The candidate prediction target (CPRED) gives the highest accuracy in both Hamming and Levenshtein distances.

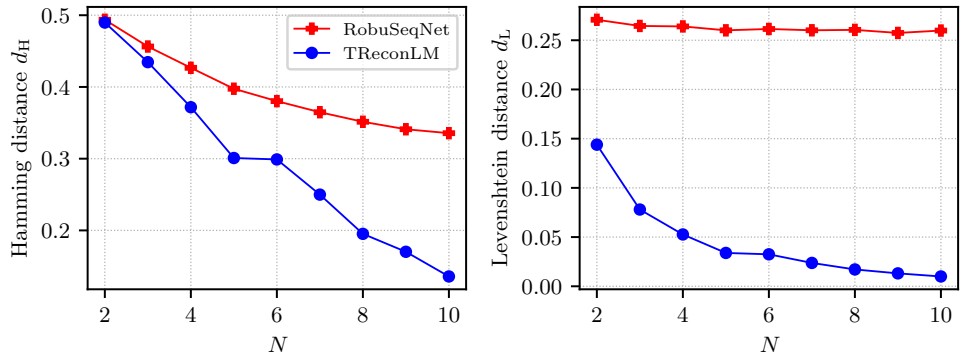

Figure 16: Comparison of TReconLM reconstruction to RobuSeqNet. TReconLM achieves lower Levenshtein distances across all cluster sizes. Our method gives overall higher positional accuracy, as the Hamming distances are lower for clusters of size four to ten and similar for two and three.

consists of an LSTM decoder with hidden dimension of 256. RobuSeqNet is trained over 32M training instances. We train small transformer models of similar size with 3M parameters also of dimension 256. As for the other experiments, we consider two models, one for two to five sequences and a second model for six to ten reads. The results are shown in Figure 16. TReconLM is able to significantly outperform RobuSeqNet even when the model size is controlled for. Here, we consider the sequence length $L = 110$.

## C.2 GPT4oMini

As an additional baseline, we consider GPT4oMini. We prompt GPT4oMini as shown in Figure 20 to perform the trace reconstruction task. We consider zero, three, and five-shot prompting. We evaluate 250 test instances of synthetic data obtained by the IDS channel for $L = 60$ and use the uniform noise distribution $\mathcal{U}[0.01; 0.1]$ for the error probabilities. Examples are generated by using the same distribution as for the test instances. For cluster sizes of two, five, and ten sequences, the Hamming and Levenshtein distances are displayed in Figure 17. We compare the performance of GPT4oMini to TReconLM by training two transformer models consisting of 20M parameters for two to five and six to ten sequences. We train on 32M instances. TReconLM outperforms GPT4oMini significantly for the trace reconstruction task in all considered cases, even though we only use a 20M parameter model.

## D PARAMETERS FOR BASELINE METHODS

Here, we provide the parameters for the baseline methods. For algorithms that utilize the error probabilities $p_I$, $p_D$, and $p_S$, we use the provided estimates when evaluating the real datasets (Antkowiak et al., 2020; Srinivasavaradhan et al., 2021). For synthetic data, we choose the mean values of the corresponding noise distribution.

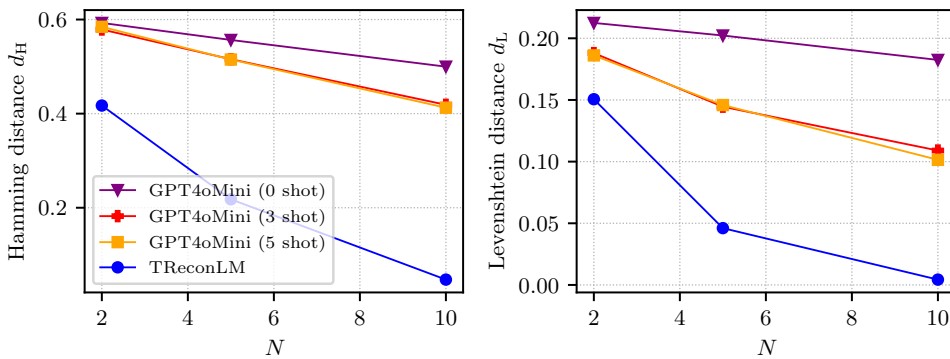

Figure 17: Comparison of GPT4oMini to TReconLM. We evaluate reconstruction using two, five, and ten noisy reads. For GPT4oMini we consider zero, three, and five-shot prompting. TReconLM is able to achieve significantly lower Hamming and Levensthein distances than GPT4oMini.

For the BMALA algorithm and VS algorithm, we use the parameters provided in Sabary et al. (2020). The BMALA method requires a window size parameter $w$, which we set to 3. The VS algorithm uses the substitution probability $p_S$ to obtain the parameter $\delta = (1+p_S)/2$. The remaining parameters are chosen as follows: $\gamma = 3/4$, $r = 2$, and $l = 3$.

The TrellisBMA algorithm uses estimates of the error probabilities. Furthermore, the algorithm requires additional parameters, which are given in Table 1 and were taken from Srinivasavaradhan et al. (2021).

| $N$ | $\beta_b$ | $\beta_e$ | $\beta_i$ |
|---|---|---|---|
| 2 and 3 | 0 | 0.1 | 0.5 |
| 4 and 5 | 0 | 1 | 0.1 |
| 6 and 7 | 0 | 0.5 | 0.1 |
| 8 and 9 | 0 | 0.5 | 0.5 |
| 10 | 0 | 0.5 | 0 |

Table 1: Parameters for TrellisBMA algorithm

# E    ATTENTION MATRIX

In order to give some interpretability of the underlying algorithm of TReconLM we visualize the attention matrices of the 20M-transformer models for both the pretrained and finetuned models. We consider the sequence length $L = 60$ and provide a heatmap of the attention matrix for prompts $p$ consisting of $N = 3$ reads. We plot the attention score of the last layer, which we obtain by performing a min-max normalization of the corresponding attention matrix values. In Figure 18 on the left, we can observe a diagonal structure, where read position $j$ attends to the sequence estimate position $j$. While earlier layers typically show a broader structure of attention scores not equal to 0, indicating the attention of multiple read positions to one sequence estimate position, the structure narrows down towards the pattern in the last layer displayed in Figure 18. The finetuned models show a similar structure, see Figure 18 right. As the Noisy-DNA dataset contains a high number of insertions towards the sequence end, we see that multiple bases in the reads attend to the last position in the sequence estimate.

# F    INCREASING THE CLUSTER SIZE

To further evaluate the potential of TReconLM, we increase the number of reads $N$ and calculate the success rate in the high noise regime on test data obtained through the IDS channel. For the test data, we consider the error characteristics as described in Section 5.2.1 where we gradually increased the lower and upper bound ($p_{LB}$ and $p_{UB}$) of the uniform noise distribution $\mathcal{U}[p_{LB}; p_{UB}]$ by 0.01 at the

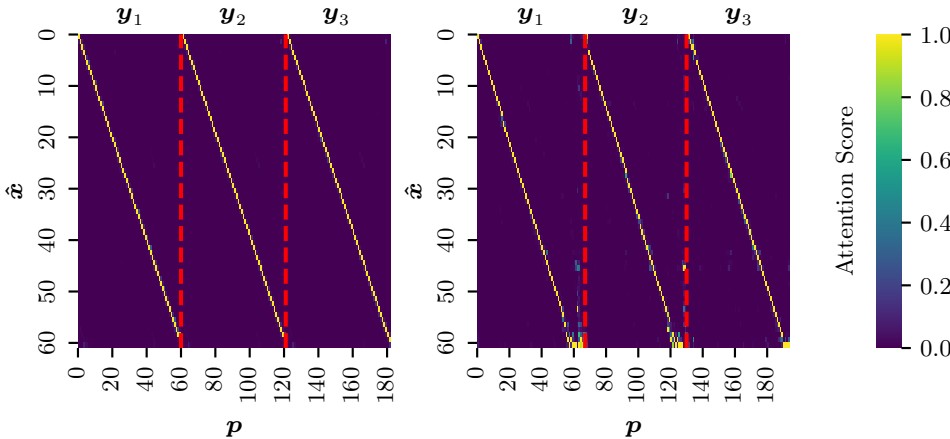

Figure 18: Visualization of the attention matrix for a prompt $p$ consisting of three reads $y_1, y_2$, and $y_3$. The red lines mark the end of reads. **Left:** Attention matrix of a pretrained model for a test example generated by the IDS channel. **Right:** Attention matrix of a finetuned model for a test instance from the Noisy-DNA dataset.

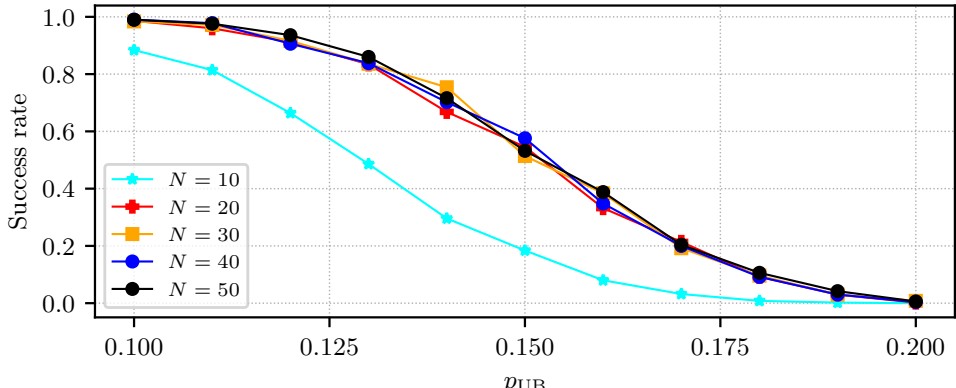

Figure 19: Success rate as a function of the upper bound of the noise distribution $p_{\mathrm{UB}}$ for larger clusters ranging from $N = 10$ to $N = 50$.

same time. We train 300M models for the reconstruction of 10, 20, 30, 40, and 50 noisy reads on the uniform noise distribution $\mathcal{U}[0.01; 0.1]$. Here, we consider the sequence length of $L = 110$ and evaluate 500 test instances. In Figure 19, we can see that increasing the number of reads from 10 to 20 gives a strong increase in success rate. Further scaling of the cluster size $N$ does not yield any benefit. However, as the models for 30, 40, and 50 reads need to be further trained, we might expect an improvement.

# G  DETAILED NUMERICAL RESULTS

For better readability and comparison, we add tables with the numbers for the following experiments from the main body: evaluation on IDS generated data for $L = 60$ (Figure 3 and Figure 4) as well as the results for the real-world data experiments: first evaluation on the Noisy-DNA dataset (Figure 6 and Figure 7) and second the Microsoft dataset (Figure 8 and Figure 9).

864
865
866
867
868
869
870
871
872
873
874
875
876
877
878
879
880
881
882
883
884
885
886
887
888
889
890
891
892
893
894
895
896
897
898
899
900
901
902
903
904
905
906
907
908
909
910
911
912
913
914
915
916
917

**Example Prompt for GPT4oMini:**

We consider a reconstruction problem of DNA sequences. We want to reconstruct a DNA sequence consisting of 60 characters (either A,C,T or G) from 5 noisy DNA sequences.
These noisy DNA sequences were generated by introducing random errors (insertion, deletion, and substitution of single characters).
The task is to provide an estimate of the ground truth DNA sequence.

Here are some examples:
Example #1
Input DNA sequences:
1. GATACGGATTGTGCTCGAGTGGATACTGGTATAGAGAAGAGAGTAATGCTAAGGTAG
2. ATATAGGACTGTTCCTCGAAGTGGATACTGTACAAAAATCAGAAGCGAGTAAGGTAG
3. GATCAGGATTGTACTCGAGTGCTACTGTACAAAGCGTCAGAGGTGCCATAGGTACG
4. GATAAAGGGACGTTGCCCGAGTGATACTGTCAAAGCGTAAAAGAGATGCTAGGTG
5. GGATCAAAGGATTGCTTGCTCGAGTGTGATACTGTACAATGATCAGAAGAGATCTAATAG
Correct output:
GATAAAGGATTGTTGCTCGAGTGGATACTGTACAAAGAGTCAGAAGAGATGCTAAGGTAG

Example #2
Input DNA sequences:
1. AAACCCTTACGGGTCGAATACATCTTATCCGAGCGCCTCAAGGAGTAGCGATTCCTAC
2. AAACCCATAGGGTCCAAAAATATTTACCGTGCACTCCGAAAGGGAGTATCGTTGATA
3. AAACACTTGGGGTCGAAAAAATACTATCCGTGTACCCCAGAGGTGTAGTGTCTCATAC
4. AACCTGAGGGTCGAAACTGTTGATCCGTGCACCTCATGAGGGTGTCGCGGCATGC
5. AAACCTTAGGGCTCGAATACATATTTACCGTGCACCTCCAGAGGAGTAGCGTTTCAA
Correct output:
AAACCCTTAGGGTCGAATACATATTTATCCGTGCACCTCCAGAGGAGTAGCGTTTCATAC

Example #3
Input DNA sequences:
1. TGCCCCGACGATATGCCGGCGGATACACTCTCACGATCGTCAAGTATATCCGTTAA
2. ATGCCCGACGCTTCTGGCCGGATACACTCAACAATCGTCACCGTTTATCCGATAA
3. ATGCCCGACGAATGCTGGCCGGATACACTTACACGATGTCAATGATATCCGAGTG
4. ATGCCCACGAGTATGCTGCCGGATCCTCACAAATCGTCAAGTTATATCCCGATAT
5. ATGCCCGATAATATATGGCGGACTCCACTCTACACGTCGTCAAGTTATATCCCGTTAG
Correct output:
ATGCCCGACGATATGCTGGCCGGATACACTCTACACGATCGTCAAGTTATATCCCGTTAT

Task:
Reconstruct the DNA sequence from the following noisy input sequences.
Input DNA sequences:
1. GGTCCCTAGAAGGATTGGATGCTGTTCGCGGGTATCTAATGTTGTGCCTTGGTGCAT
2. AGGTCGCCCAGAAGTGATATGGTCGCTGGTCGCGGCATCTAATGTTGTGACATCTTGAT
3. AGGTTACCCTGATAGTGATGTAGTGTGCATTTCGCGGCTCTATGTTGTGCCTGTTGCT
4. AGGTCCTAGTAAGGTATATGCATGCGGTCGCGGCTCTAATGTTGTGCTTGAGTTGCT
5. AGCTCCGTAGAGGAATGATGCTGTTCGCCGGCATTAGATGTGTGCCTCGGTTGCT
Provide an estimate of the ground truth DNA sequence consisting of 60 characters in the
format ***estimated DNA sequence*** - use three * on each side of the estimated DNA sequence.

Figure 20: Example of a three-shot prompt for GPT4oMini.

| Success rate | | | | | | | |
|---|---|---|---|---|---|---|---|
| $N$ | VS | RobuSeqNet | MUSCLE | TrellisBMA | BMALA | ITR | TReconLM |
| 2 | 2e-4 | 6e-4 | 2e-4 | 0 | 2e-4 | 1.8e-3 | 6e-3 |
| 3 | 1.8e-3 | 1.08e-2 | 4.4e-3 | 0 | 2.08-2 | 5.2e-3 | 0.121 |
| 4 | 2.2e-3 | 1.94e-2 | 2.42e-2 | 4.86e-2 | 5.36-2 | 0.157 | 0.311 |
| 5 | 3.6e-3 | 2.96e-2 | 5.5e-2 | 0.103 | 0.113 | 0.321 | 0.516 |
| 6 | 3.8e-3 | 3.66e-2 | 7.28e-2 | 0.12 | 0.170 | 0.509 | 0.648 |
| 7 | 2.8e-3 | 5.08e-2 | 9.06e-2 | 0.214 | 0.245 | 0.665 | 0.77 |
| 8 | 7.4e-3 | 6.52e-2 | 0.117 | 0.292 | 0.322 | 0.775 | 0.862 |
| 9 | 9.8e-3 | 7.80e-2 | 0.164 | 0.373 | 0.377 | 0.848 | 0.914 |
| 10 | 8.4e-3 | 8.16e-2 | 0.173 | 0.461 | 0.423 | 0.885 | 0.942 |
| Hamming distance $d_{\mathrm{H}}$ | | | | | | | |
| $N$ | VS | RobuSeqNet | MUSCLE | TrellisBMA | BMALA | ITR | TReconLM |
| 2 | 0.566 | 0.427 | 0.57 | 0.481 | 0.485 | 0.508 | 0.395 |
| 3 | 0.548 | 0.371 | 0.58 | 0.443 | 0.386 | 0.671 | 0.258 |
| 4 | 0.534 | 0.336 | 0.482 | 0.31 | 0.327 | 0.331 | 0.161 |
| 5 | 0.525 | 0.305 | 0.443 | 0.249 | 0.273 | 0.245 | 9.68e-2 |
| 6 | 0.522 | 0.28 | 0.439 | 0.231 | 0.230 | 0.138 | 6.82e-2 |
| 7 | 0.518 | 0.258 | 0.435 | 0.180 | 0.192 | 9.08e-2 | 4.24e-2 |
| 8 | 0.500 | 0.237 | 0.408 | 0.135 | 0.161 | 5.69e-2 | 2.46e-2 |
| 9 | 0.499 | 0.224 | 0.377 | 0.106 | 0.130 | 3.89e-2 | 1.615-2 |
| 10 | 0.498 | 0.214 | 0.366 | 8.81e-2 | 0.124 | 2.95e-2 | 9.64e-3 |
| Levenshtein distance $d_{\mathrm{L}}$ | | | | | | | |
| $N$ | VS | RobuSeqNet | MUSCLE | TrellisBMA | BMALA | ITR | TReconLM |
| 2 | 0.159 | 0.224 | 0.205 | 0.316 | 0.223 | 0.152 | 0.14 |
| 3 | 0.159 | 0.201 | 0.149 | 0.294 | 0.154 | 0.241 | 6.28e-2 |
| 4 | 0.174 | 0.191 | 0.106 | 0.124 | 0.122 | 6.63e-2 | 3.61e-2 |
| 5 | 0.162 | 0.183 | 8.13e-2 | 8.92e-2 | 9.81e-2 | 3.85e-2 | 2.04e-2 |
| 6 | 0.16 | 0.174 | 7.34e-2 | 9.28e-2 | 7.88e-2 | 2.02e-2 | 1.35e-2 |
| 7 | 0.159 | 0.167 | 6.70e-2 | 6.78e-2 | 6.23e-2 | 1.19e-2 | 7.86e-3 |
| 8 | 0.160 | 0.158 | 5.72e-2 | 4.92e-2 | 5.09e-2 | 6.9e-3 | 4.47e-3 |
| 9 | 0.164 | 0.154 | 4.87e-2 | 3.86e-2 | 4.27e-2 | 4.44e-3 | 2.67e-3 |
| 10 | 0.165 | 0.153 | 4.56e-2 | 2.99e-2 | 3.73e-2 | 3.07e-3 | 1.71e-3 |

Table 2: Results for IDS data with $L = 60$ (see Figure 3 and Figure 4).

| | | | Success rate | | | |
|---|---|---|---|---|---|---|
| $N$ | VS | MUSCLE | BMALA | ITR | TReconLM (p) | TReconLM (f) |
| 2 | 2.05e-2 | 3.03e-2 | 3.26e-4 | 1.89e-2 | 1.95e-3 | 0.174 |
| 3 | 2.08e-2 | 6.28-2 | 1.30e-3 | 4.92e-2 | 1.53e-2 | 0.424 |
| 4 | 2.51e-2 | 0.135 | 4.23e-3 | 0.203 | 2.86e-2 | 0.556 |
| 5 | 2.12e-2 | 0.223 | 2.94e-3 | 0.395 | 4.33e-2 | 0.627 |
| 6 | 4.25e-2 | 0.359 | 6.57e-3 | 0.467 | 9.30e-2 | 0.761 |
| 7 | 3.54e-2 | 0.415 | 7.58e-3 | 0.581 | 0.101 | 0.794 |
| 8 | 4.10e-2 | 0.41 | 7.08e-3 | 0.56 | 0.155 | 0.817 |
| 9 | 3.99e-2 | 0.474 | 6.07e-3 | 0.628 | 0.208 | 0.835 |
| 10 | 2.73e-2 | 0.509 | 7.58e-3 | 0.61 | 0.353 | 0.843 |
| | | | Hamming distance $d_{\mathrm{H}}$ | | | |
| $N$ | VS | MUSCLE | BMALA | ITR | TReconLM (p) | TReconLM (f) |
| 2 | 0.476 | 0.426 | 0.503 | 0.449 | 0.5 | 0.192 |
| 3 | 0.463 | 0.43 | 0.446 | 0.505 | 0.363 | 0.126 |
| 4 | 0.441 | 0.337 | 0.42 | 0.289 | 0.29 | 9.16e-2 |
| 5 | 0.442 | 0.276 | 0.396 | 0.182 | 0.244 | 7.34e-2 |
| 6 | 0.405 | 0.186 | 0.364 | 0.145 | 0.176 | 5.62e-2 |
| 7 | 0.409 | 0.17 | 0.356 | 9.79e-2 | 0.162 | 4.08e-2 |
| 8 | 0.391 | 0.177 | 0.354 | 0.109 | 0.145 | 3.32e-2 |
| 9 | 0.396 | 0.137 | 0.350 | 8.28e-2 | 0.129 | 2.74e-2 |
| 10 | 0.402 | 0.116 | 0.344 | 8.75e-2 | 0.117 | 2.43e-2 |
| | | | Levenshtein distance $d_{\mathrm{L}}$ | | | |
| $N$ | VS | MUSCLE | BMALA | ITR | TReconLM (p) | TReconLM(f) |
| 2 | 8.97e-2 | 0.11 | 0.161 | 8.53e-2 | 8.67e-2 | 4.52e-2 |
| 3 | 9.5e-2 | 0.101 | 0.12 | 0.112 | 6.04e-2 | 2.74e-2 |
| 4 | 0.12 | 7.54e-2 | 0.116 | 5.84e-2 | 5.17e-2 | 2.08e-2 |
| 5 | 0.107 | 6.91e-2 | 0.107 | 5.65e-2 | 5.27e-2 | 1.84e-2 |
| 6 | 9.52e-2 | 6.12e-2 | 9.09e-2 | 4.13e-2 | 3.64e-2 | 7.78e-3 |
| 7 | 9.92e-2 | 6.29e-2 | 8.55e-2 | 4.03e-2 | 3.52e-2 | 6.77e-3 |
| 8 | 0.108 | 5.84e-2 | 8.54e-2 | 3.80e-2 | 3.24e-2 | 5.90e-3 |
| 9 | 0.111 | 5.51e-2 | 8.34e-2 | 3.69e-2 | 3.04e-2 | 5.49e-3 |
| 10 | 0.116 | 5.53e-2 | 8.26e-2 | 3.58e-2 | 2.53e-2 | 5.21e-3 |

Table 3: Results for Noisy-DNA dataset (see Figure 6 and Figure 7). Pretrained models (p) and finetuned models (f).

| | Success rate | | | | | | |
|---|---|---|---|---|---|---|---|
| $N$ | VS | MUSCLE | BMALA | TrellisBMA | ITR | TReconLM (p) | TReconLM (f) |
| 2 | 1.87e-2 | 6.69e-3 | 4.23e-3 | 7.05e-4 | 3.74e-2 | 3.28-e2 | 0.402 |
| 3 | 1.84e-2 | 1.65e-2 | 9.33e-2 | 1.36e-2 | 1.47e-2 | 0.361 | 0.735 |
| 4 | 2.32e-2 | 0.252 | 0.146 | 0.349 | 0.555 | 0.599 | 0.871 |
| 5 | 3.14e-2 | 0.419 | 0.296 | 0.528 | 0.675 | 0.756 | 0.924 |
| 6 | 3.25e-2 | 0.432 | 0.378 | 0.635 | 0.8 | 0.84 | 0.903 |
| 7 | 3.33e-2 | 0.433 | 0.492 | 0.697 | 0.841 | 0.887 | 0.933 |
| 8 | 4.88e-2 | 0.634 | 0.55 | 0.754 | 0.858 | 0.918 | 0.959 |
| 9 | 4.45e-2 | 0.72 | 0.624 | 0.797 | 0.873 | 0.935 | 0.968 |
| 10 | 4.76e-2 | 0.735 | 0.672 | 0.823 | 0.876 | 0.944 | 0.975 |

| | Hamming distance $d_{\mathrm{H}}$ | | | | | | |
|---|---|---|---|---|---|---|---|
| $N$ | VS | MUSCLE | BMALA | TrellisBMA | ITR | TReconLM (p) | TReconLM (f) |
| 2 | 0.464 | 0.49 | 0.387 | 0.342 | 0.408 | 0.337 | 0.223 |
| 3 | 0.456 | 0.555 | 0.237 | 0.265 | 0.63 | 0.165 | 0.103 |
| 4 | 0.405 | 0.253 | 0.184 | 0.111 | 9.87e-2 | 8.69e-2 | 5.73e-2 |
| 5 | 0.412 | 0.199 | 0.126 | 7.95e-2 | 9.37e-2 | 5.07e-2 | 3.95e-2 |
| 6 | 0.408 | 0.209 | 0.105 | 5.02e-2 | 5.17e-2 | 3.38e-2 | 2.01e-2 |
| 7 | 0.408 | 0.232 | 8.3e-2 | 4.17e-2 | 4.33e-2 | 2.44e-2 | 1.30e-2 |
| 8 | 0.365 | 0.13 | 7.15e-2 | 3.46e-2 | 4.22e-2 | 1.87e-2 | 8.02e-3 |
| 9 | 0.365 | 9.86e-2 | 5.66e-2 | 2.83e-2 | 3.75e-2 | 1.55e-2 | 6.33e-3 |
| 10 | 0.364 | 9.25e-2 | 4.99e-2 | 2.60e-2 | 3.91e-2 | 1.34e-2 | 5.24e-3 |

| | Levenshtein distance $d_{\mathrm{L}}$ | | | | | | |
|---|---|---|---|---|---|---|---|
| $N$ | VS | MUSCLE | BMALA | TrellisBMA | ITR | TReconLM (p) | TReconLM (f) |
| 2 | 5.68e-2 | 6.69e-2 | 0.119 | 0.154 | 5.29e-2 | 5.26e-2 | 2.06e-2 |
| 3 | 5.67e-2 | 4.85e-2 | 5.18e-2 | 0.119 | 9.16e-2 | 1.56e-2 | 6.09e-3 |
| 4 | 7.23e-2 | 1.56e-2 | 4.01e-2 | 0.0163 | 8.56e-3 | 7.78e-3 | 2.61e-3 |
| 5 | 6.08e-2 | 9.38e-3 | 2.59e-2 | 1.01e-2 | 4.87e-3 | 4.34e-3 | 1.45e-3 |
| 6 | 5.96e-2 | 8.78e-3 | 2.06e-2 | 7.19e-3 | 2.64e-3 | 2.68-3 | 1.58e-3 |
| 7 | 6.23e-2 | 8.75e-3 | 1.48e-2 | 5.63e-3 | 1.98e-3 | 1.87e-3 | 1.01e-3 |
| 8 | 6.51e-2 | 4.62e-3 | 1.22e-2 | 4.62e-3 | 1.67e-3 | 1.33e-3 | 5.87e-4 |
| 9 | 6.63e-2 | 3.27e-3 | 9.55e-3 | 3.66e-3 | 1.45e-3 | 1.09e-3 | 4.62e-4 |
| 10 | 6.87e-2 | 3.10e-3 | 8.06e-3 | 3.33e-3 | 1.38e-3 | 9.22e-4 | 3.78e-4 |

Table 4: Results for Microsoft dataset (see Figure 8 Figure 9). Pretrained models (p) and finetuned models (f).

