# OpenReview forum: "Trace Reconstruction for DNA Data Storage using Language Models"
_ICLR.cc/2025/Conference — Submitted to ICLR 2025_

### Official Review · Reviewer_GUws · 2024-10-26

**Soundness:** 3
**Presentation:** 3
**Contribution:** 3
**Rating:** 5
**Confidence:** 4

**Summary:**

This paper explores trace reconstruction of DNA sequences using a GPT model. Trace reconstruction aims to recover the original DNA sequence from several corrupted versions containing insertion, deletion, and substitution errors. On synthetic data with a fixed error probability, the GPT model performs well. However, while a pretrained GPT model did not achieve effective results on real data, fine-tuning significantly improved its performance, surpassing previous methods.

**Strengths:**

Applying GPT to DNA storage problems is a promising direction, especially since tasks like multiple sequence alignment and trace reconstruction typically demand high computational resources. This work suggests the potential of using large language models (LLMs) to streamline these processes.

**Weaknesses:**

Unlike traditional dynamic programming-based algorithms, the proposed GPT model relies heavily on dataset-specific statistics, as seen in real-world experiments where fine-tuning was necessary. Could training GPT with various error probabilities enable broader applicability, including for real data?
Section 5.3.2 shows that fine-tuning does not perform well on larger cluster sizes. Could the authors elaborate on the reasons for this limitation?
Including data and code would enable more thorough validation and replication of the results.

**Questions:**

Please check weaknesses

---

> ### Author Response · Authors · 2024-11-28
> **Response to reviewer GUws**
>
> The authors thank the reviewer for the feedback. In the following we address the concerns and questions in the order as pointed out by the reviewer.
>
> - The reviewer noted that “unlike traditional dynamic programming-based algorithms, the proposed GPT model relies heavily on dataset-specific statistics, as seen in real-world experiments where fine-tuning was necessary’'. Interestingly, our proposed method generalizes relatively well under a distribution shift. Specifically, we see that a model pretrained on the IDS channel and evaluated on the Microsoft dataset, which is distributed differently than the training distribution (different error probabilities and the sequences exhibit long-range dependencies and are therefore not uniform at random as is pretraining data), still gives similar and often better performance than state-of-the-art reconstruction algorithms like ITR. Finally, in practice finetuning is usually possible since in DNA data storage the error distributions for a given system are either known or can be measured.
>
> - Regarding “Could training GPT with various error probabilities enable broader applicability, including for real data?” We saw that training on the uniform distribution $\mathcal{U}[0.01;0.1]$ achieves good performance. Training directly on the estimated noise values of the Microsoft did not perform better than training on $\mathcal{U}[0.01;0.1]$. Therefore, we believe that our suggested training generation mechanism is already a good choice,  it covers the error probabilities of most DNA data storage systems. Our method already outperforms existing methods on real data by a wide margin.
>
> - The issue with the performance drop for larger clusters in Section 5.3.2 (Figure 9) could be resolved with additional hyperparameter tuning. We updated the plots accordingly.
>
> - Regarding missing data and code. We uploaded the code and datasets, including all hyperparameters for each experiment as well as reproduction instructions.
>
> We hope the new experiments and clarifications as well as making the code available address the reviewer’s concerns, and we would greatly appreciate reconsideration of the score. We are happy to provide further clarifications if needed.

---

> > ### Comment · Reviewer_GUws · 2024-11-30
> >
> > Thank you for the detailed response and updates. While I appreciate the effort in clarifying the questions and providing the data and code, the explanation regarding the performance drop for larger clusters and the generalization capabilities of the proposed method remains insufficiently convincing. Additionally, the claim of broad applicability is not fully substantiated by the experiments provided. Therefore, I will maintain my original score.

---

### Official Review · Reviewer_be5j · 2024-10-28

**Soundness:** 2
**Presentation:** 2
**Contribution:** 2
**Rating:** 3
**Confidence:** 4

**Summary:**

The author(s) introduced a DL-based approach for reconstructing DNA sequences from clustered DNA reads obtained through a DNA storage pipeline.  The study utilizes a next-token prediction transformer for this purpose. The model is pre-trained on synthetic data and subsequently fine-tuned using a specific open wet-lab dataset.

**Strengths:**

Sequence reconstruction is a challenging and valuable topic within the fields of DNA storage and bioinformatics.

The introduction of a next-token prediction transformer-based method for sequence reconstruction marks a novel contribution to the DNA storage research community.

The experimental outcomes are promising.

**Weaknesses:**

The reviewer’s primary concern relates to the limited novelty within the machine learning or learning representation community, as the work appears to be an application of established deep learning techniques. The authors may spend more context to address this comment in their rebuttal if there is one.

The writting was somehow not treated carefully. For example, since the caption of Figure 3, the proposed method is abruptly referred to as “GPT”. Firstly, the proposed method lacks a formal name, and “GPT” is not a good choice. Secondly, even GPT is not predifined in the text.

**Questions:**

1. Line 199. The author(s) assert that they "do not consider this (the discrepancy between the pretraining and finetuning data) here" without providing a rationale. The reviewer thought this may not be acceptable.
1. As a DL-based method, the absence of comparative experiments with the closely related transformer models, DNAformer and RobuSeqNet, within the main text requires a justifiable explanation.
1. Why is it necessary to train separate models for (sequence length $N=2$ to $N=5$) and (sequence length $N=6$ to $N=10$)? If this is necessary, the sequence length is a hyperparameter that requires analysis. (In Line 420, the author(s) refer to "sequences of length two to five". The reviewer suspects that this might actually refer to the cardinality of clusters, with the range extending from $N=2$ to $N=5$.)
1. What is the "subcluster" for in Line 392?
1. The reviewer posits that no method is capable of effectively reconstructing sequences from clusters with a cardinality of  $N=2$, due to the insufficient information available. However, the finetuned results presented in Figures 6 and 8 for $N=2$ are promising and appear to contradict this assertion. The authors are kindly requested to provide an explanation for this. Could it be that the model learned underlying patterns or distributions specific to the wet-lab data?

---

> ### Author Response · Authors · 2024-11-28
> **Response to reviewer be5j**
>
> We thank the reviewer for their comments and feedback.
>
> Weaknesses:
> The reviewer's main concern is novelty. The novelty of our proposed methods lies in proposing next-token prediction as an approach to trace reconstruction and showing that it achieves state-of-the-art performance. Yes, next-token prediction is widely used in ML, however it is not obvious at all that this paradigm enables state-of-the-art performance for trace reconstruction. This is testified by the fact that there are several other prior works that propose deep learning methods for trace reconstruction (such as DNAformer and RobuSeqNet) that rely on other ideas and perform significantly worse as demonstrated in our paper. Regarding the naming of the proposed method, we updated our draft with a formal name (TReconLM which stands for Trace Reconstruction with a Language Model) for readability and revised throughout the paper to improve readability.
>
> Questions:
> 1. We reformulated the sentence on Line 199 to increase clarity. We do in fact have experiments where we study the performance when the train and test distribution differ significantly in the finetuning experiments, and show that the corresponding performance drop can be overcome with finetuning.
>
> 2. The main body now contains comparisons to both RobuSeqNet and DNAformer.
>
> 3. It is beneficial to train separate models, specifically we saw that training separate models for each value of the cluster size N gives the best performance. Since this is prohibitively expensive in terms of training compute we chose to train only two models instead of nine, one for each N from two to ten. Regarding “In Line 420, the author(s) refer to "sequences of length two to five". The reviewer suspects that this might actually refer to the cardinality of clusters, with the range extending from N=2 to N=5.” Yes, thank you for pointing that out. We updated the draft accordingly.
>
> 4. A subcluster is a cluster split further into subclusters. Real-world data may contain clusters that have more than 10 sequences. We split these into subclusters to process these larger clusters with our method.
>
> 5. We believe that by finetuning we can adapt to the data-specific errors, e.g., high number of insertions towards the sequence end in the Noisy-DNA dataset, as well as learning dataset-specific patterns, e.g., the Microsoft dataset contains long-range dependencies. This explains the significant difference between the finetuned models and the baseline methods on real-world data. However, the success rates in Figure 6 and 8 for N = 2 are 0.174 and 0.402, which is far from perfect reconstruction.
>
> We hope the new experiments and clarifications address the reviewer’s concerns, and we would greatly appreciate reconsideration of the score. We are happy to provide further clarifications if needed.

---

> > ### Comment · Reviewer_be5j · 2024-11-28
> >
> > I would like to thank the authors for their clarifications and the additional experiments.
> >
> > However, my concern about the novelty in ML of the work remains. Applying standard methods to a new task may not provide sufficient novelty for a top-tier ML conference like ICLR. This work might be more appropriately suited for a specialized bioinformatics conference.
> >
> > I choose to keep my score.

---

### Official Review · Reviewer_7P2t · 2024-10-30

**Soundness:** 4
**Presentation:** 4
**Contribution:** 4
**Rating:** 8
**Confidence:** 4

**Summary:**

The paper presents a method for trace reconstruction in DNA data storage using language models. The approach involves training on synthetic data, with subsequent finetuning on real datasets to improve performance on technology-specific error distributions. Experimental results show that the proposed method outperforms existing approaches on the Hamming and Levenshtein distance metrics.

**Strengths:**

- The paper is very well written.
- The approach is innovative.
- The concept is novel to my knowledge.
- The use case (high noise, few traces) is very relevant to real-world scenarios.
- Superior performance compared to the state of the art is demonstrated.

**Weaknesses:**

Major:
- Although the model can handle different error patterns through fine-tuning, it may struggle with entirely new or highly variable error profiles in unseen data. Specifically, it is unclear how well burst errors would be handled.

Minor:
- Figure 3: The parameter N (number of traces) should be introduced.
- Line 235: Please explain why you are considering decoder-only transformers.

**Questions:**

- One-hot encoded sequences are padded to a fixed predetermined length. As DNA synthesis methods are being developed that produce significantly longer oligonucleotides, is this fixed-length scheme generally compatible with sequencing technologies that produce long variable-length reads (nanopore)?
- The authors assume that the original sequence consists of bases chosen uniformly at random. If fine-tuning on real data would not be feasible, how would the scheme need to be adapted to accommodate non-uniform distributions?
- It would be nice to see an evaluation of the minimum number of traces required for given I/D/S error probabilities.

---

> ### Author Response · Authors · 2024-11-28
> **Response to reviewer 7P2t**
>
> Many thanks for the feedback and the positive evaluation of our work!
>
> Major weakness:
>
> - We agree that the model might struggle with data that is far from the training data. From the finetuning experiments in Figure 7, we see that without finetuning, when the data is relatively far from the training data, ITR sometimes performs slightly better than our method, but with finetuning on the target distribution our method TReconLM significantly outperforms ITR and all other baselines. For DNA data storage this is not an issue, since there we can adapt to the distribution with finetuning.
>
> - Regarding burst errors, they can be handled to some extent. The reads of the Noisy-DNA dataset show a high value of “C” and “T” towards the sequence end. The results show that our reconstruction method is able to handle this type of error.
>
> Minor weaknesses:
>
> - Thanks for the suggestion, we fixed that.
> - We also considered other sequence-to-sequence models, specifically state-space models based on the Mamba architecture, however transformers worked better so we proceeded with them.
>
> Questions:
>
> - Regarding compatibility with nanopore, yes the method is compatible with nanopore reads. In Section 5.3.2 the data (Microsoft data) is from nanopore and with that data it works well. Regarding longer reads, at the moment the synthesis technologies that scale well do all not give sequences of length more than about 200, thus in the near and medium future the reads will remain relatively short. We think, however that the method also scales to larger read lengths and plan to add experiments on that. In our work, padding of the reads is not necessary, and we work directly with the nucleotides without one-hot-encoding, but one-hot-encoding with padding has been successfully applied by Bar-Lev et al., 2024.
>
> - The method is to some extent robust to a shift in training and test distribution (see Finetuning experiments in Section 5.3). The Microsoft dataset contains sequences that exhibit long-range dependencies and are therefore not uniform at random. Still pretrained transformer models can perform similarly and often better compared to state-of-the-art reconstruction algorithms like ITR. When finetuning on real data is not possible but we have knowledge about the distribution it is typically possible to generate finetuning data.
>
> - We are working on an experiment to find the minimum number of traces for a given set of error probabilities. As these experiments are still running, we are unfortunately not yet able to give a quantitative answer to this question. However, in the new Appendix F we study a variant of this question.

---

> > ### Comment · Reviewer_7P2t · 2024-11-28
> >
> > Thank you for your answers to my comments. An evaluation of the minimum number of traces required for given I/D/S error probabilities in the main body of the paper would significantly add to its value. I maintain my score.

---

### Official Review · Reviewer_YysM · 2024-11-02

**Soundness:** 3
**Presentation:** 3
**Contribution:** 3
**Rating:** 6
**Confidence:** 5

**Summary:**

The authors propose a new LLM-based method for the trace reconstruction problem. The problem is reconstruct an unknown string given N noisy copies of the string. These noisy copies ("reads") are a natural byproduct of DNA sequencing technologies. They contain insertion, deletion, and substitution errors. To date, there is no know algorithm to solve this problem theoretically or empirically. For real systems, people use either combinatorial or network based algorithms to output the unknown string from a collection of 2 to 10 noisy reads. The authors compare their GPT-architecture solution to these other algorithms.

For a success metric, the authors use either perfect reconstruction fraction, or measure the average edit/Hamming distances of the reconstructed strings from the ground truth strings. They compare two versions of the GPT-based model: (1) pre-trained on purely synthetic data, (2) fine-tuned on real data. The difference in (2) is that real data may follow a different error pattern, as opposed to synthetic data in (1) which has uniformly distributed errors.

Compared to other methods, the authors show improved performance. In the appendix, they also compare a few other versions, including models with fewer parameters.

**Strengths:**

The authors provide a new method that gives good results for trace reconstruction, for both real and synthetic datasets.

The authors justify that fine-tuning is both necessary and sufficient to adapt to real data error distributions.

The authors compare against several existing trace reconstruction algorithms, giving a thorough picture of the current ways to solve the problem in practice.

The paper is concise and easy to follow. The plots are clear, with consistent coloring and informative captions.

**Weaknesses:**

The paper showcases a successful use of LLMs to solve a real problem. But the solution is not particularly surprising, especially as written. Transformers are very powerful, and the trace reconstruction inputs are consistently structured, without a very long context, so the model is clearly going to do pretty well if trained on the exact same data.

I think the biggest open question that is not answered by this paper is: How does the GPT model solve the trace reconstruction problem? Is there a way to analyze the underlying "algorithm" to get some insights into why it does better than the baselines? Are there specific types of instances where GPT does better than ITR consistently? The results between GPT and ITR are pretty close in many of the graphs (e.g., N > 7), so it would be nice to know concretely why / how GPT is outperforming ITR.

I would love to see some more insights that may generalize to other statistical problems. For example, was there anything the authors learned about pre-training or fine-tuning? Are there any lessons about the training algorithm? Or does everything just work "out of the box" with standard implementations of small GPT models? The success of the authors' method may inspire future efforts on using LLMs to solve statistical problems, and therefore, it would be good to share as detailed insights as possible. For example, one open question is what is the minimum model size needs for the trace reconstruction setting in this paper? It seems like 300M is enough, but 3M is too small?

The paper is missing some key technical details about the training process, hyperparameter tuning, training time, number of GPUs, etc. I would like to see this in the final paper for full transparency.

**Questions:**

1) Another baseline would be how well can GPT Mini / Gemini Flash solve this problem in a few-shot manner? Is there a prompt that can get similar performance without needing to pre-train or fine-tune? This would make it easier to implement via an API rather than needing GPUs and knowing how to train LLMs (e.g, for biologists who want to solve this problem).

2) Can you perform some interpretability of the network to see what algorithm the GPT model is implementing under the hood? And do you think the fine-tuned model is doing something fundamentally different than the pre-trained model?

3) It would be helpful to dig deeper into this topic, and push the limits of what the LLMs are capable of. The point here is not just to try random other settings (this would not be a good use of time). The point is to develop general insights that can guide future researchers that want to know if an LLM can solve their (more complicated) statistical problem. For example, there are many variations of this problem that are easy to try synthetically (or others that the authors think would help undercover ideas that could inform (2) above):
- use very long reads, e.g., O(1000) bases --> do you need a much bigger model for this?
- increasing the read count and the error rate -- what happens if p_UB goes up to 0.4 but there are O(100) reads per cluster?
- for Figure 5, how many reads are necessary to get to ~0.0 error?

4) Overall, I would be happy to increase my score if the authors could show some more general findings beyond just "LLMs can solve trace reconstruction if they have enough data and enough parameters"

Minor comments:
- the logarithmic plot y-axis is a bit hard to read, I would also add into the appendix a table of the results for the average Hamming / Levenshtein distances at each N.
- Typo in Appendix D title "PARAMTERS"

-----------------

Post rebuttal: Increasing score 5 --> 6 since the authors added new parts to the paper, including:
1. attention maps
2. more training details
3. scaling laws
4. large cluster experiments

I encourage the authors to incorporate these more detailed / technical analyses to the main body of the paper, to make it clear that their method took some care to get to work.

---

> ### Author Response · Authors · 2024-11-28
> **Response to reviewer YysM**
>
> We thank the reviewer for the detailed feedback and insightful comments.
>
> Regarding the Weaknesses:
>
> - Regarding “The paper showcases a successful use of LLMs to solve a real problem. But the solution is not particularly surprising, especially as written.’’ We found it remarkable that the method works significantly better on a real-world problem of importance and found it not entirely expected given that there are several competing trace-reconstruction methods, including neural network-based methods (like DNAformer and RobuSeqNet) that perform significantly worse.
>
> - In order to better understand how our GPT-based method, which we now termed TReconLM for readability, solves the trace reconstruction better than competing methods, we carried out the following additional experiments:
>
> - In Appendix E, we plot the attention maps of the last layer and find that typically, when estimating a given position, the attention maps attend to the right position in the traces.
>
> - To investigate why TReconLM does better than baselines like ITR we analyzed problem instances and found that for easy problem instances where the traces are close, ITR and TReconLM perform equally well, but on harder problem instances where the traces are further away, TReconLM performs better.
>
> - Regarding important insights on what makes the method work: There are several details that one needs to get right for the method to work well, for example varying the error probabilities during the pretraining stage as we do through sampling the error probabilities $p_\mathrm{I}$, $p_\mathrm{D}$, and $p_\mathrm{S}$ from a uniform distribution is important. In particular, a model trained on the uniform distribution $\mathcal{U}[0.01;0.1]$ for the error probabilities and tested on 0.05 performs better than a model only trained on 0.05. As another example, we also tried chain-of-thought-like targets (see Appendix B), that did not perform well, so directly predicting the sequence estimate works well.
>
> - Thanks for the suggestion to study the impact of model size further. We added a new section with preliminary scaling laws (Section 5.4) where we study the performance of transformers ranging from 3M to 300M parameters trained up to 32M problem instances. We see from the preliminary scaling plot that further increasing the number of training problem instances will likely give further benefits. The corresponding experiments are still running and will be completed in a week or two. From the current results, it can already be seen how the method improves as we invest more compute, and that moderately sized transformers like 80M ones can do very well.
>
> - Regarding key technical details: We uploaded our code, including all hyperparameters for each experiment and run times of the GPUs during the rebuttal, and we included the training details in the paper.
>
> Questions:
> 1. We added a comparison to GPT4oMini as a baseline in Appendix C.2 with zero, three, and five-shot evaluation on synthetic data. It can be seen that our method gives the best result in the considered range.
>
> 2. In order to provide some interpretability of our method we added visualizations of the attention matrices in the last layer of the transformer models of both pretrained and finetuned models.
>
> 3. As suggested, in order to push the limits of LLMs we increase the number of reads and evaluate on high noise values ($p_\mathrm{UB}=[0.1;0.2]$ see Section 5.2.1). However, the models for this experiment require further training.
>
> 4. Thanks for being open to changing your score. We think it is remarkable that TReconLM (a GPT) performs so well at trace reconstruction, which is not entirely expected given that several prior deep learning methods perform worse (RobuSeqNet and DNAformer). Moreover, it is critical to get several details right, like how to formulate the next-word prediction, and how exactly to generate training instances.
>
> Minor comments:
>
> We thank the reviewer for pointing out the typo. We added tables for the Hamming and Levenshtein distances as well as success rates in the Appendix G for an exact comparison of the results.

---

> > ### Comment · Reviewer_YysM · 2024-11-29
> >
> > Thanks for the updated comments. I think the new experiments and new additions to the appendix add a lot of value to the paper. I am happy to increase my score 5 --> 6. I still feel like that paper falls into the category of "Transformers are very powerful and can solve many problems" which is borderline for a conference like ICLR. Nonetheless, the contribution is novel and well-supported.
> >
> > For the next version of the paper, I would suggest that the authors find a way to fit these two important observations into the main body of the paper:
> > - "We found it remarkable that the method works significantly better on a real-world problem of importance and found it not entirely expected given that there are several competing trace-reconstruction methods, including neural network-based methods (like DNAformer and RobuSeqNet) that perform significantly worse." --> this is great motivation, and makes the authors' contributions more interesting because it is surprising in these ways
> > - "Regarding important insights on what makes the method work: There are several details that one needs to get right for the method to work well" --> I would work these into section 4, since this is essentially part of the training recipe that you are proposing for TReconLM
> > - "To investigate why TReconLM does better than baselines like ITR we analyzed problem instances and found that for easy problem instances where the traces are close, ITR and TReconLM perform equally well, but on harder problem instances where the traces are further away, TReconLM performs better." --> I would also that this could be a nice qualitative subsection in the main paper in Section 5

---

### Meta-Review · Area_Chair_h692 · 2024-12-21

**Metareview:**

**Summary:**
This paper proposes a novel LLM-based method for the trace reconstruction problem, which is the problem of estimating the original DNA sequence from a number $N$ of noisy copies of it, where the noise is introduced via the DNA synthesis and sequencing processes. This paper models the noising process as the insertion-deletion-substitution (IDS) channel, and proposes training of an LLM to perform next-word prediction with training instances of the form given in equation (2).

**Strengths:**
The reviewers evaluated the proposal as innovative and promising. Most of them also valued the experimental results which exhibited superior empirical performance of the proposal compared with several baselines.

**Weaknesses:**
- As some reviewers mentioned, the idea would be regarded as a straightforward application of the idea of next-word prediction in LLMs to the trace reconstruction problem.
- This paper does not discuss the issue of contaminated sequences (Qin et al., 2024) or complementary strands, the latter of which could efficiently be dealt with the Siamese architecture (Bar-Lev et al., 2024).

**Reasons:**
The proposal of applying the idea of next-word prediction to the trace reconstruction would seem rather straightforward, so that the audience of the confefence might not be able to learn much out of it, even though it would be efficient in solving this particular problem.

Minor points:
- Abstract: next-token prediction → next-word prediction
- In the description of the IDS channel, how an output sequence $y$ is generated from an input sequence $x$ is not well described. There is no mention in Figure 2 to the output $y_\ell$. I guess that in "substitution" of $x_\ell$ one chooses one of the three letters that are different from $x_\ell$ *with equal probabilities*, but it is not described explicitly.
- I think that it would be more appropriate to look at the failure rate rather than the success rate, which would allow the authors to demonstrate more clearly how their proposal outperforms ITR.
- I did not understand the experimental setting in Section 5.2.1. I guess that the parameters $p_I,p_D,p_S$ are randomly selected from the uniform distribution $\mathcal{U}(0.01+0.01k,0.1+0.01k)$ with $k\in\{0,\ldots,10\}$, and these values are then used in generating test samples.

**Additional Comments On Reviewer Discussion:**

The review scores exhibited a relatively large split. I would agree with the authors in that it is not obvious at all to show that the next-word prediction idea would be efficient in the trace reconstruction problem. Nevertheless, I feel, similarly to Reviewer GUws, that one can learn very little from the findings provided by this paper, beyond the empirical efficiency of the proposal to this particular problem. It does not seem either that the proposal would be a significant breakthrough in the relevant literature, although it may suggest a promising approach.

---

### Decision · Program_Chairs · 2025-01-22

Reject